# Overcoming barriers to the registration of new plant varieties under the DUS system

Chin Jian Yang [1], Joanne Russell[2], Luke Ramsay[2], William Thomas[2], Wayne Powell[1] & Ian Mackay [1,3✉]

Distinctness, Uniformity and Stability (DUS) is an intellectual property system introduced in 1961 by the International Union for the Protection of New Varieties of Plants (UPOV) for safeguarding the investment and rewarding innovation in developing new plant varieties. Despite the rapid advancement in our understanding of crop biology over the past 60 years, the DUS system has changed little and is still largely dependent upon a set of morphological traits for testing candidate varieties. As the demand for more plant varieties increases, the barriers to registration of new varieties become more acute and thus require urgent review to the system. To highlight the challenges and remedies in the current system, we evaluated a comprehensive panel of 805 UK barley varieties that span the entire history of DUS testing. Our findings reveal the system deficiencies such as inconsistencies in DUS traits across environments, limitations in DUS trait combinatorial space, and inadequacies in currently available DUS markers. We advocate the concept of genomic DUS and provide evidence for a shift towards a robust genomics-enabled registration system for new crop varieties.

[1] Scotland's Rural College (SRUC), Kings Buildings, West Mains Road, Edinburgh EH9 3JG, UK. [2] The James Hutton Institute, Invergowrie, Dundee DD2 5DA, UK. [3] IMplant Consultancy Ltd., Chelmsford, UK. ✉email: i.j.mackay@gmail.com

Crop breeding involves a considerable investment of time, resources, and money by seed companies to produce improved varieties of plants. Plant Variety Rights (PVR) is a form of intellectual property introduced in 1961 by the International Union for the Protection of New Varieties of Plants (UPOV) to protect the breeders' investment in creating new varieties, support innovation, and serve market demand[1]. For almost 60 years, the protection of new plant varieties through the award of PVR relied on passing two tests: Distinctness, Uniformity, and Stability (DUS), and Value for Cultivation and Use (VCU). DUS requires the new variety to be distinct from the common knowledge varieties, uniform across seeds that constitute the variety, and stable across environments[2]. DUS is usually defined by a set of morphological traits, although isozyme electrophoresis and molecular markers are occasionally used[3]. On the other hand, VCU requires the new crop variety to demonstrate improvement in yield, biotic or abiotic resistance, and quality characteristics[4]. Unlike VCU traits such as yield and disease resistance that have been the center of attention in crop breeding[5], the DUS system has received relatively little attention despite its pivotal role in the registration of new varieties[6,7].

The pressure on the current DUS system stems from multiple issues. As more new varieties arise, the DUS trait combinatorial space becomes more limited and requires additional effort in breeding unique DUS trait combinations. Many DUS traits have low heritabilities[7] which means more trait variability due to environmental fluctuations and limited reliability of DUS trait scores outside of the trial environment. While the current system is well established for major crops, it is hard to implement in minor or orphan crops since the traits for DUS are hard to determine[8,9]. Furthermore, the current DUS system is largely designed for inbred species or varieties which is hardly practical in outbreeding species or hybrid varieties[10]. Lastly, the current definitions of new varieties[2] and essentially derived varieties[11] rely on a fine line concerning which characteristics are considered essential, which complicates an objective evaluation of a candidate variety[12,13].

Over the years, many attempts at improving the DUS system have met with little success. Suggestion for the use of molecular markers in DUS traces back to at least 1990 using minisatellites in soft fruits[14]. Since then, more molecular markers have been proposed for DUS, for example, 28 SSR markers in maize[15], 25 SNP markers in barley[16], and 5 SSR markers in rice[17]. However, the number of available DUS markers that have been proposed thus far is too few and low throughput. More recently, larger marker sets using SNP arrays have been suggested, including 3,072 SNP markers in maize[18] and 6,000 SNP markers in soybean[19]. As of now, none of these have been officially adopted by the UPOV. Instead, UPOV currently requires the use of molecular markers only when they correlate with the DUS traits perfectly[20], which does not reflect the advances in genotyping technologies and understanding of DUS trait genetics.

Using the UK barley DUS system as a test case (panel of 805 spring, winter, and alternative barley varieties that have been accepted into the UK national list (NL), as well as 28 DUS traits), we demonstrate both the challenges and opportunities for the creation of a new DUS system. We show that the current DUS system is lacking in consistencies across the environment, limited in trait combinatorial space, and impaired by a non-optimal marker system. We suggest the idea of genomic DUS for overcoming various issues in the current DUS system and demonstrate its advantages in plant variety registration.

## Results and discussion

**DUS trait and marker data**. The 28 barley DUS traits include the seasonal type and 27 above-ground morphologies, including leaves, ears, and spikelets (Table 1). Currently, within the UK, barley DUS trait data are publicly available from the National Institute of Agricultural Botany (NIAB) in England and the Science and Advice for Scottish Agriculture (SASA) in Scotland. We obtained the data from these two sources and supplemented it with additional data from Cockram et al.[7]. The NIAB data serves as our primary data as it is more complete than the SASA data, which was only used for comparative analysis. 21 DUS traits are scored on a scale of 1 to 9 or a smaller subset of the scale, and seven traits are scored on a binary scale, all of which were based on criteria defined in Supplementary Data 1. Of the 27 traits excluding seasonal type, two traits are not segregating in spring barley and one in winter barley (Supplementary Fig. 1). The missing rate in the DUS trait data ranges from 0 to 78%, with only 5 traits above 10%. In addition, our analysis included marker data for 805 varieties from the IMPROMALT collection (http://www.barleyhub.org/projects/impromalt/), of which 710 had DUS trait data.

**DUS trait inconsistencies across environments**. Comparison across DUS trait scoring organizations (NIAB vs. SASA) showed an overall consistency in two-thirds of the DUS trait scores (Fig. 1a and b). For each variety, the consistency was measured as the proportion of DUS traits that are exact matches between the two organizations, and the overall consistency was derived from the means of all variety consistencies. In most cases, the trait score differences within each variety are small (mean = 0.55, sd = 0.28, $n = 395$, two-sided t-test $p < 0.05$) (Fig. 1a, Supplementary Fig. 2). These differences are expected given that the DUS traits were scored in different environments by different DUS inspectors. There is little to no bias in trait score differences between NIAB and SASA (Supplementary Fig. 2) except for trait 6 (flag leaf: glaucosity of the sheath) and trait 25 (grain: spiculation of inner lateral nerves of the dorsal side of lemma). On average, trait 6 is about 1 score higher in NIAB compared to SASA while trait 25 is about 1 score lower in NIAB, which may reflect the environmental effects on these traits. Regardless, with the reduction in DUS trait combinatorial space as measured by shrinkage in DUS trait Manhattan distances over time (Fig. 1c & d), especially in spring barley, small trait score differences can easily complicate variety identification. Manhattan distances are the sums of absolute differences between any two variables and lower distances imply reduced variation in DUS traits among the compared varieties. This may risk some barley varieties failing DUS testing in one country but not another due to variations in DUS traits. Besides, the inconsistencies are present in the majority (392/395) of the barley varieties compared, which suggests that the inconsistencies are common and excludes the possibility of poor data handling by either organization. Given the roles of the DUS system in granting PVR, a two-third consistency across organizations is inadequate and risky.

Of all 28 barley DUS traits, 15 have low heritabilities ($h^2 < 0.50$) (Table 1, Supplementary Table 1) and are thus contradictory for DUS purposes. As previously defined by Falconer and Mackay[21], heritability is a measure of "degree of correspondence between phenotypic value and breeding value". Therefore, for any given variety, traits with low heritabilities have little replicability in trait values obtained from different environments (e.g. year, location). As expected, the DUS trait inconsistencies across scoring organizations are negatively correlated (−0.67) with heritabilities (Fig. 1e). Similar results were observed when the heritabilities were calculated from spring and winter barley separately. Consequently, instead of a fair evaluation of the genetic merits underlying new varieties, the current DUS system simply determines new varieties based on environmental stochasticity.

**Table 1 DUS trait names and heritabilities, standard errors in parentheses.**

| Trait | Name | $h^2$ | | |
| --- | --- | --- | --- | --- |
| | | Combined | Spring | Winter |
| 1 | Kernel: colour of the aleurone layer | 0.78 (0.04) | 0.16 (0.06) | 0.79 (0.06) |
| 2 | Plant: growth habit | 0.25 (0.05) | 0.17 (0.06) | 0.24 (0.07) |
| 3 | Lowest leaves: hairiness of leaf sheaths | 0.75 (0.04) | NA | 0.69 (0.07) |
| 4 | Flag leaf: intensity of anthocyanin colouration of auricles | 0.74 (0.05) | 0.19 (0.06) | 0.84 (0.08) |
| 5 | Flag leaf: attitude | 0.28 (0.13) | 0.28 (0.19) | 0.25 (0.16) |
| 6 | Flag leaf: glaucosity of sheath | 0.12 (0.04) | 0.05 (0.03) | 0.10 (0.05) |
| 7 | Time of ear emergence (first spikelet visible on 50% of ears) | 0.28 (0.05) | 0.20 (0.06) | 0.26 (0.07) |
| 8 | Awns: intensity of anthocyanin colouration of tips | 0.67 (0.05) | 0.09 (0.04) | 0.83 (0.08) |
| 9 | Ear: glaucosity | 0.42 (0.05) | 0.45 (0.07) | 0.33 (0.08) |
| 10 | Ear: attitude | 0.25 (0.05) | 0.26 (0.07) | 0.17 (0.06) |
| 11 | Plant: length (stem, ear and awns) | 0.17 (0.04) | 0.13 (0.05) | 0.14 (0.06) |
| 12 | Ear: number of rows | 1.00 (0.01) | NA | 1.00 (0.03) |
| 13 | Ear: shape | 0.10 (0.04) | 0.04 (0.03) | 0.09 (0.05) |
| 14 | Ear: density | 0.23 (0.05) | 0.14 (0.05) | 0.24 (0.07) |
| 15 | Ear: length (excluding awns) | 0.18 (0.05) | 0.05 (0.04) | 0.29 (0.08) |
| 16 | Awn: length (compared to ear) | 0.18 (0.04) | 0.15 (0.05) | 0.11 (0.05) |
| 17 | Rachis: length of first segment | 0.34 (0.05) | 0.32 (0.07) | 0.28 (0.07) |
| 18 | Rachis: curvature of first segment | 0.26 (0.05) | 0.25 (0.07) | 0.20 (0.07) |
| 19 | Ear: development of sterile spikelets | 1.00 (0.04) | 1.00 (0.06) | 1.00 (0.09) |
| 20 | Sterile spikelets: attitude (in mid-third of ear) | 0.64 (0.06) | 0.63 (0.08) | 0.49 (0.10) |
| 21 | Median spikelet: length of glume and its awn relative to grain | 0.15 (0.04) | 0.07 (0.04) | 0.18 (0.06) |
| 22 | Grain: rachilla hair type | 1.00 (0.01) | 1.00 (0.02) | 0.84 (0.05) |
| 23 | Grain: husk | 0.01 (0.02) | 0.04 (0.03) | 0.00 (0.02) |
| 24 | Grain: anthocyanin colouration of nerves of lemma | 0.69 (0.05) | 0.31 (0.07) | 0.78 (0.07) |
| 25 | Grain: speculation of inner lateral nerves of dorsal side of lemma | 0.74 (0.04) | 0.49 (0.07) | 0.78 (0.06) |
| 26 | Grain: hairiness of ventral furrow | 0.96 (0.02) | 0.65 (0.07) | 0.94 (0.04) |
| 27 | Grain: disposition of lodicules | 0.91 (0.02) | 0.99 (0.03) | NA |
| 28 | Seasonal type | 1.00 (0.00) | NA | NA |

**Limitations to DUS trait combinatorial space due to genetic gain in yield.** 12 out of 21 barley DUS traits have non-zero genetic correlations with yield in spring barley (Fig. 2), which risk undesirable correlated responses upon selecting for either DUS traits or yield. Non-zero genetic correlations are the hidden cost in exchange for the genetic gain in yield. In traits with low phenotypic correlations, the unintended selection for DUS traits may not be immediately apparent to breeders. For instance, both DUS traits 10 (ear: attitude) and 11 (plant: length) are negatively correlated with yield, which translates to semi-dwarf barley plants with erect ears having a higher yield than tall barley plants with recurved ears. Such correlations could help define ideal crop ideotypes[22], however, they are not ideal for DUS purposes because high-yielding plants are more likely to be semi-dwarf with erect ears. As the genetic gain in yield increases over time[23], it is inevitable that DUS trait combinatorial space gets more limited (Fig. 1c and d) due to correlated selection responses. On the other hand, selection away from DUS trait combinatorial space risks losing the genetic gain in yield. While we have only considered correlations between DUS traits and yield, there are other VCU traits that may also constrict DUS trait combinatorial space.

**Flaws in the current DUS marker system in capturing complex trait genetic architecture.** GWAS results showed that 14 of 28 barley DUS traits are likely regulated by few major loci and some of these loci are likely fixed in either spring or winter barley populations (Table 2). Of the total 32 GWAS loci, 30 were identified in the combined dataset (Supplementary Table 2, Supplementary Fig. 3), 12 in the spring-only dataset (Supplementary Table 3, Supplementary Fig. 4), and 16 in the winter-only dataset (Supplementary Table 4, Supplementary Fig. 5). Part of the explanation for the difference is due to the individual datasets having a smaller sample size and thus lower power. Another reason is that some traits are not segregating or are rare in either spring or winter germplasm. Examples of these traits are: 3 (lowest leaves: hairiness of leaf sheaths), 12 (ear: number of rows), 23 (grain: husk), 26 (grain: hairiness of ventral furrow), and 27 (grain: disposition of lodicules). A major QTL for trait 3 is tightly linked to *Vrn-H2*, a major vernalization locus[24] while traits 12, 23, and 27 are largely monomorphic in the UK barley breeding pool due to preferences for two-rowed barley with hulled grains and clasping (collar type) lodicules. In comparison with previous work on DUS traits GWAS[7], the number of loci increased from 16 to 32 with 12 loci in common.

In accordance with the UPOV guidelines[20], molecular markers can only be used in DUS if they confer a direct relationship with the DUS traits. This might work well with those 14 traits with known major loci, although there is a risk of ignoring effects from minor or exotic loci. One such example would be anthocyanin-related traits in flag leaf (trait 4) and awn (trait 8), where *anthocyaninless 1* (*ant1*) and *ant2* are segregating in winter but not spring barley varieties in the UK (Table 2). Unless the DUS markers for *ant1* and *ant2* are in perfect linkage with the causative polymorphisms, these markers would give misleading results if used in spring barley. To complicate this issue further, we identified a locus at *ant2* for an anthocyanin-related trait in grain (trait 24) in spring barley, which may suggest an additionally linked locus that is segregating in spring barley responsible for grain-only anthocyanin pigmentation. On the other hand, it is improbable to create molecular markers that would tag any of the other 14 traits without major loci.

To extend beyond locus-specific markers, a small marker set for DUS has been proposed[25] although our evaluation showed limited distinguishing power. By simulating $F_6$ progeny from known parent pairs, we compared the marker set from these

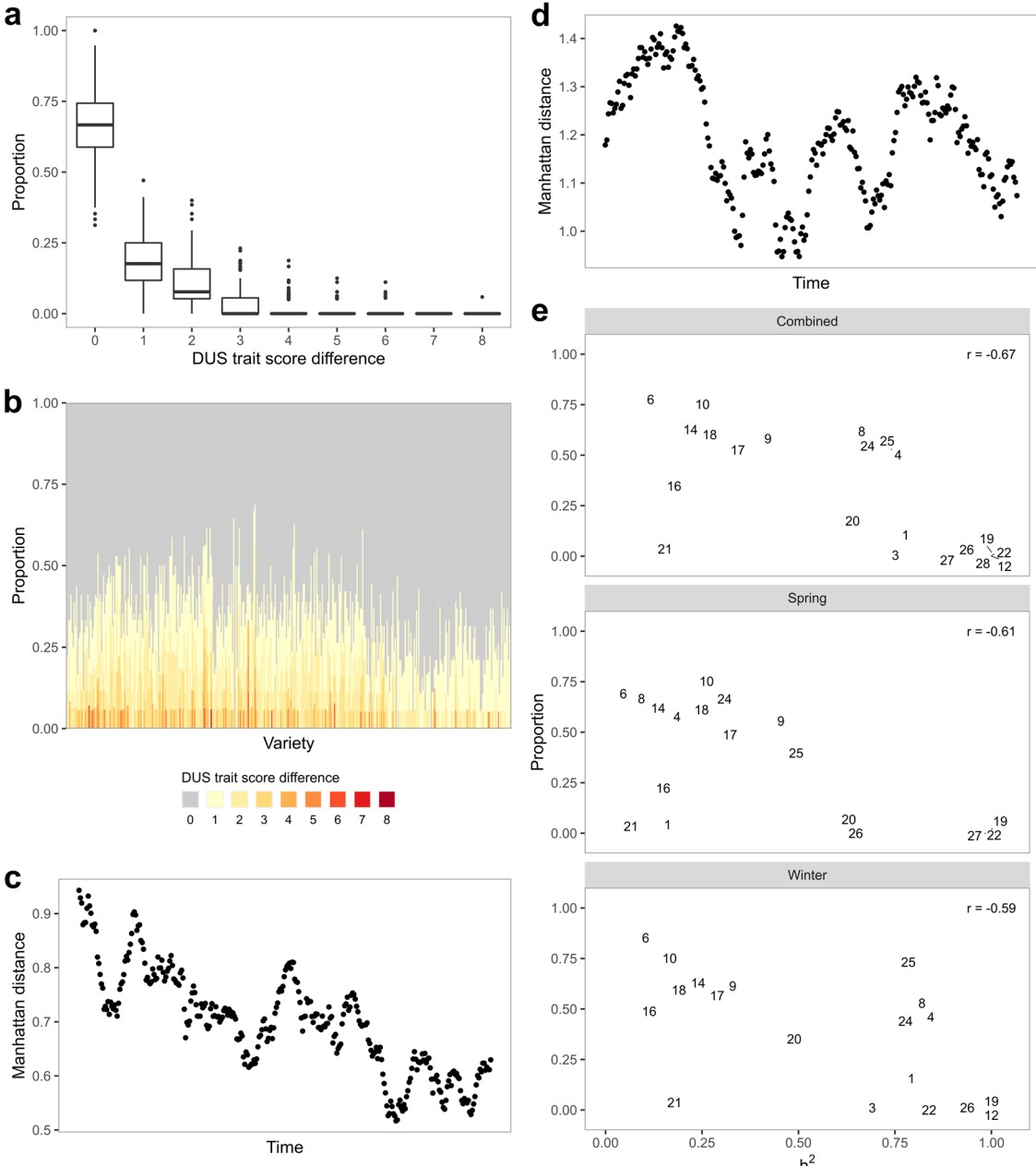

**Fig. 1 DUS trait discrepancies and combinatorial space. a** Boxplots of the proportion of DUS trait score differences between NIAB and SASA data. Center line indicates median, lower and upper hinges indicate first and third quartile, while lower and upper whisker indicates 1.5 times the interquartile range from the hinges. **b** Proportion of DUS trait score differences for each variety, with the oldest variety (1963) on the left and the newest variety (2007) on the right. **c** Rolling mean distances of 20 spring barley varieties calculated from DUS traits with an increment of one new variety at a time. The leftmost point on the "Time" axis indicates the mean from the 20 earliest varieties, while the rightmost point indicates the mean from the 20 latest varieties. **d** Rolling mean distances of 20 winter barley varieties calculated from DUS traits with an increment of one new variety at a time. **e** Relationships between the proportion of DUS trait score differences and heritabilities, separated by all (spring and winter combined), spring only and winter only groups. Each point is shown as its trait number, which is available in Table 1.

simulated progeny to their parents, actual variety (progeny of the parent pairs), and other simulated progeny. While most of these simulated progeny remained unique in older varieties, this is not true for newer varieties (Fig. 3a and b, Supplementary Data 2), especially in spring barley. For example, LG Goddess matched perfectly with 7.5% of the simulated progeny, and its parents Octavia and Shada matched perfectly with 8.0% and 7.8% of the simulated progeny, respectively (Supplementary Data 2). Furthermore, 88.4% of the simulated progeny have over 1% probability of matching with other simulated progeny (Supplementary Data 2).

A small marker set for DUS is problematic in a crop in which genomic diversity progressively gets narrower over time. Of the total 39 markers[25], only 4 to 22 markers are segregating between the parents analyzed. Besides, these markers are not randomly distributed as there are some in strong linkage disequilibrium (LD) which would not be informative.

As a follow-up, we investigated the number of markers required for proper separation of varieties in DUS and determined that approximately 500–1000 markers are likely the minimum (Fig. 4a). By comparing the Manhattan distances

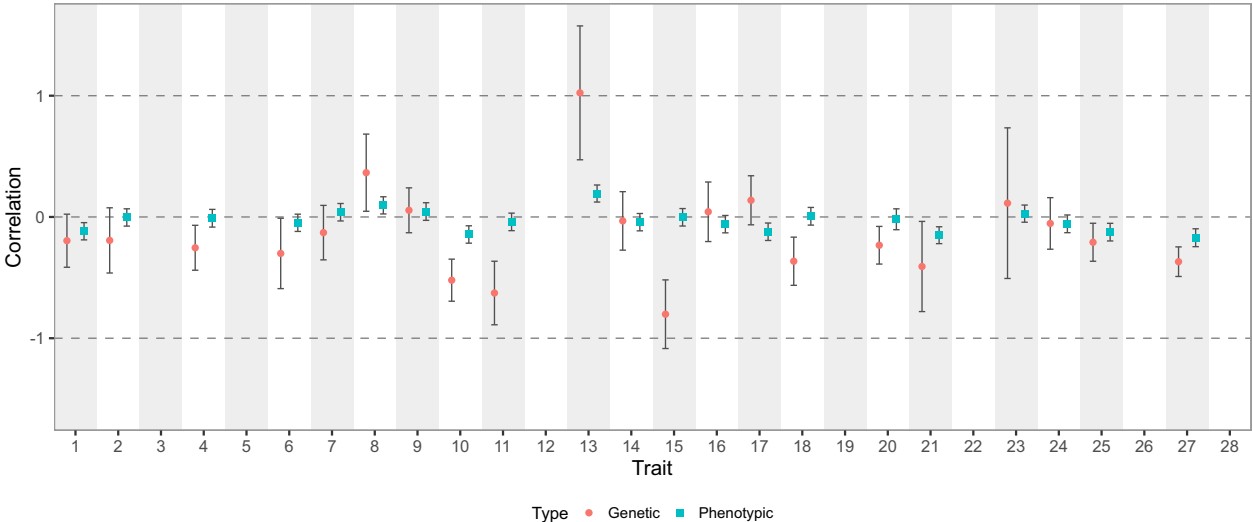

**Fig. 2 Genetic and phenotypic correlations between DUS traits and yield.** Genetic and phenotypic correlations, along with their standard errors, are shown for each DUS trait and yield. No correlation is available for traits 3, 5, 12, 19, 22, 26, and 28 because of either a high missing rate or lack of variation in the DUS trait.

### Table 2 GWAS results.

| Trait | GWAS peak[a] | | | | | Allele frequency | | | Known gene/locus | |
|---|---|---|---|---|---|---|---|---|---|---|
| | Chr | Pos (Mb) | Effect | $-\log_{10}p$ | Pop[b] | C | S | W | Name | Pos (Mb) |
| 1 | 4H | 525.07 | 0.74 | 96.02 | C,S,W | 0.19 | 0.02 | 0.37 | MbHF35[c] (*Blx1*)[45] | 534.04 |
| 2 | 3H | 631.83 | −0.69 | 6.40 | C | 0.49 | 0.05 | 0.98 | HORVU3Hr1G090910[d,46] | 633.53 |
| 3 | 1H | 473.27 | −0.25 | 4.92 | C | 0.16 | 0.01 | 0.32 | NA | NA |
| 3 | 4H | 631.68 | 3.32 | 116.93 | C,W | 0.44 | 0.00 | 0.91 | HORVU4Hr1G085920[d], HORVU4Hr1G085590[d] (*Hsh1*)[24] | 633.03 |
| 4 | 2H | 676.76 | −2.25 | 42.77 | C,W | 0.25 | 0.00 | 0.53 | HORVU2Hr1G096810 (*Ant2*)[7] | 676.85 |
| 4 | 7H | 73.55 | −0.80 | 10.67 | C | 0.10 | 0.02 | 0.19 | HORVU7Hr1G034630 (*Ant1*)[47] | 72.92 |
| 8 | 2H | 675.76 | −2.27 | 55.97 | C,W | 0.25 | 0.00 | 0.53 | HORVU2Hr1G096810 (*Ant2*)[7] | 676.85 |
| 8 | 6H | 536.07 | 0.57 | 6.75 | S | 0.30 | 0.16 | 0.45 | NA | NA |
| 8 | 7H | 73.55 | −0.69 | 11.01 | C,W | 0.10 | 0.02 | 0.19 | HORVU7Hr1G034630 (*Ant1*)[47] | 72.92 |
| 9 | 1H | 0.29 | −0.52 | 6.23 | C,S,W | 0.07 | 0.07 | 0.06 | EAR-G_1[48] | 0.50[e] |
| 9 | 2H | 6.18 | −0.33 | 6.98 | C | 0.43 | 0.52 | 0.34 | NA | NA |
| 11 | 4H | 608.43 | −0.30 | 5.93 | C | 0.45 | 0.49 | 0.41 | NA | NA |
| 12 | 2H | 663.88 | 0.04 | 6.05 | C,W | 0.30 | 0.24 | 0.37 | HORVU2Hr1G092290 (*Vrs1*)[49] | 651.03 |
| 12 | 5H | 579.73 | −0.04 | 6.11 | C | 0.42 | 0.00 | 0.89 | HORVU5Hr1G081450 (*Vrs2*)[50] | 564.41 |
| 13 | 3H | 437.24 | 0.63 | 5.56 | C | 0.05 | 0.00 | 0.11 | NA | NA |
| 15 | 4H | 608.38 | −0.22 | 4.76 | S | 0.45 | 0.49 | 0.41 | 4_5[51] | 618.00[e] |
| 19 | 2H | 652.42 | −0.49 | 146.59 | C,S,W | 0.29 | 0.24 | 0.34 | HORVU2Hr1G092290 (*Vrs1*)[52] | 651.03 |
| 20 | 1H | 404.92 | −0.38 | 11.39 | C,S,W | 0.30 | 0.02 | 0.60 | HORVU1Hr1G051010 (*Vrs3*)[53] | 378.41 |
| 20 | 2H | 655.81 | −0.63 | 19.19 | C,S,W | 0.22 | 0.20 | 0.25 | HORVU2Hr1G092290 (*Vrs1*)[52] | 651.03 |
| 20 | 3H | 659.54 | −0.19 | 4.12 | C | 0.16 | 0.01 | 0.32 | NA | NA |
| 20 | 5H | 488.46 | −0.16 | 4.37 | C | 0.06 | 0.02 | 0.11 | NA | NA |
| 21 | 7H | 47.56 | 0.17 | 7.93 | C,W | 0.08 | 0.07 | 0.08 | NA | NA |
| 22 | 5H | 542.50 | −0.16 | 24.72 | C,S,W | 0.36 | 0.52 | 0.18 | *Srh*[7] | 547.24[e] |
| 23 | 7H | 612.52 | −0.31 | 6.71 | C,S | 0.06 | 0.10 | 0.02 | HORVU7Hr1G089930 (*Nud*)[54] | 546.59 |
| 24 | 2H | 676.20 | −1.45 | 44.08 | C,S,W | 0.34 | 0.14 | 0.56 | HORVU2Hr1G096810 (*Ant2*)[7] | 676.85 |
| 24 | 7H | 72.97 | −0.56 | 8.46 | C,W | 0.22 | 0.03 | 0.43 | HORVU7Hr1G034630 (*Ant1*)[47] | 72.92 |
| 25 | 2H | 638.37 | 1.97 | 57.54 | C,S,W | 0.17 | 0.09 | 0.26 | *Gth1*[7] | 647.46[e] |
| 26 | 6H | 0.33 | 3.85 | 152.61 | C,W | 0.14 | 0.00 | 0.29 | 11_20881[7] | 5.20[e] |
| 27 | 2H | 724.71 | −0.10 | 24.45 | C,S | 0.48 | 0.05 | 0.95 | HORVU2Hr1G113880 (*Cly1*)[55] | 730.03 |
| 28 | 1H | 511.92 | −0.51 | 47.74 | C | 0.48 | 0.00 | 1.00 | HORVU1Hr1G076430 (*Ppd-H2*)[56] | 514.1 |
| 28 | 4H | 643.68 | −0.63 | 63.95 | C | 0.48 | 0.00 | 1.00 | *Vrn-H2*[57] | NA |
| 28 | 5H | 571.03 | −0.34 | 38.29 | C | 0.46 | 0.00 | 0.98 | HORVU5Hr1G095630 (*Vrn-H1*)[58] | 599.09 |

Significant GWAS peaks (FDR < 0.05) are summarized here along with their closest known gene or locus.
[a]If the GWAS peak is found in more than one population, only the results from the combined (C) analysis are shown here.
[b]This column indicates which populations (C: Combined, S: Spring, W: Winter) showed significance for any given GWAS peak.
[c]MbHF35 is a cluster of 3 linked genes: HvMYB4H (HORVU4Hr1G063760), HvMYC4H (NA), and HvF35H (HORVU4Hr1G063780).
[d]Unverified candidate genes.
[e]Approximated physical positions based on genetic positions.

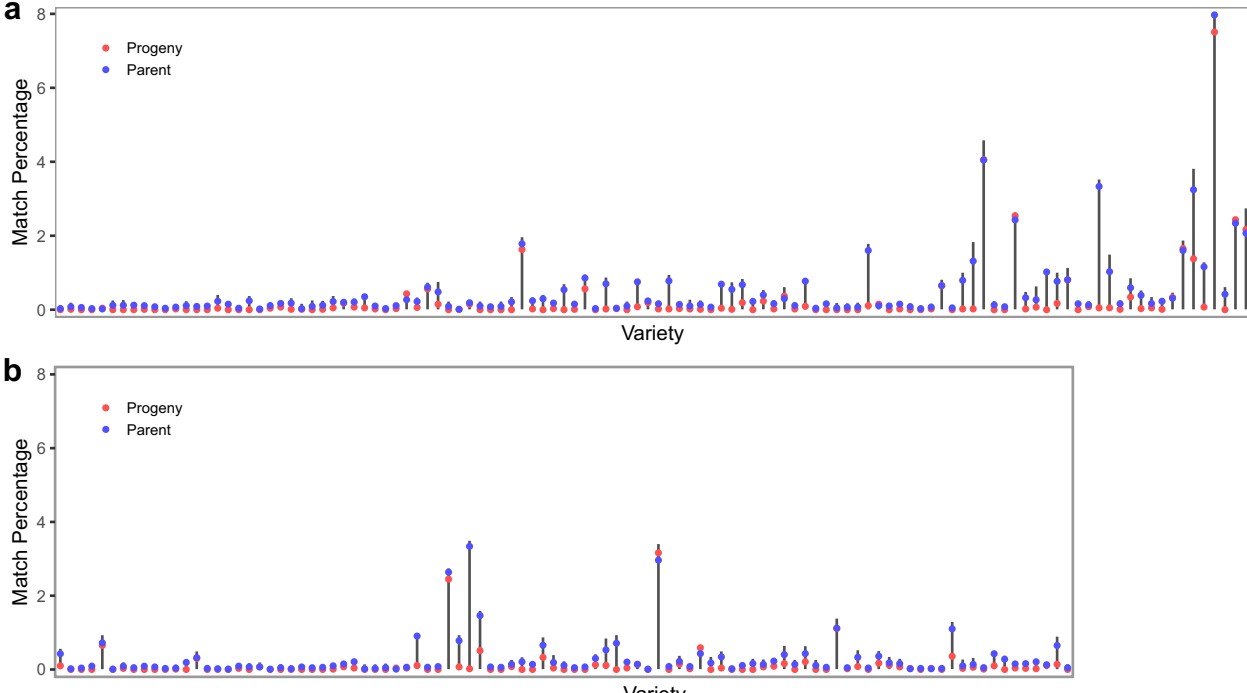

**Fig. 3 Usefulness of a small marker set for DUS.** This small marker set is taken from 39 DUS markers in Owens et al.[21]. Percentage of exact matches in marker data between simulated $F_6$ progeny and their real sibling and parents. Grey vertical bars represent the spread of match percentage among the simulated progeny. **a** 114 spring barley varieties sorted from oldest (left, 1982) to newest (right, 2016). **b** 97 winter barley varieties sorted from oldest (left, 1987) to newest (right, 2016).

calculated from all 28 DUS traits against a series of randomly sampled markers, the correlation between these two distances begins to plateau at about 500–1000 markers. The correlation maxes out at about 0.60, which is similar to the value previously observed by Jones et al.[10]. This is not surprising given that the correlation depends on the DUS trait heritabilities. Manhattan distances determined from DUS traits with high heritabilities ($h^2 > 0.50$) showed a stronger correlation with Manhattan distances from the marker data than DUS traits with low heritabilities ($h^2 < 0.50$) (Fig. 4a). In addition, the distribution variances stabilize at a similar range too (Supplementary Table 5), which affirms that any marker set smaller than 500 markers is insufficient.

**Genomic DUS: concept and practices.** Given the various issues we have described in the DUS system so far, the remaining option is to use genomic markers. There are multiple ways to implement genomic markers in DUS, and we will provide a simple example here using Manhattan distances, which is one of many measures of dissimilarities among varieties. Under haploid marker coding of 0 and 1, the Manhattan distance between any two varieties is equivalent to $2 \times (1 - similarity)$ where *similarity* is measured as the proportion of exact marker matches between two varieties. Similar to the current DUS system, we will need a reference panel (common knowledge varieties set) and the genomic marker data for the reference panel. As an example, we set all 805 barley varieties as our reference panel and computed the Manhattan distances among these varieties. The distances are divided by within and across seasonal types, as the values ranged from 0.04 to 0.69 within spring barley, 0.04 to 0.87 within winter barley, and 0.44 to 0.97 between spring and winter barley (Fig. 4b). To demonstrate how genomic markers work in DUS, we simulated 1000 $F_6$ and $BC_1S_4$ progeny from two pairs of parents in spring barley. The first parent pair is Propino and Quench, which has a distance of 0.20 and thus represents the "low" distance between

parents. The second parent pair is Riviera and Cooper, which has a distance of 0.59 and thus represents the "high" distance parents. Given an arbitrary minimum threshold of 0.05 for distinctness, 13.0% of $F_6$ progeny and 59.6% of $BC_1S_4$ progeny from the low parents would be rejected for lack of distinctness, while none of the $F_6$ progeny and 4.9% of the $BC_1S_4$ progeny from the high parents would be rejected (Fig. 4c).

Another important consequence of using genomic markers in DUS is the regulation of essentially derived varieties (EDVs)[11]. As of the current standard, the definition of EDVs is unclear[12] and it often involves complicated and expensive court proceedings to determine EDVs[13]. Furthermore, the information on whether a market variety is an EDV is not available in any of the current literature, and it is possible that no EDV ever makes it into the market. With genomic markers, any varieties submitted for DUS evaluation that failed to pass the minimum distance threshold would be considered for EDVs. Curiously, among the varieties in our reference panel, four varieties did not pass our arbitrary minimum threshold of 0.05 (Fig. 4b). Spring barley Class, and winter barley KWS Joy, Mackie, and Angora all had distances of 0.04 with their previously submitted parents Prestige, Wintmalt and KWS Tower and full sib Melanie, respectively. Since only 4 out of a total of 326,836 pairwise comparisons had a distance below the minimum threshold, it is not possible to visualize them in Fig. 4b. In addition, of these 4 pairs, Angora and Melanie were previously deemed indistinguishable in their DUS traits and had to be separated by either microsatellite markers[26] or electrophoresis of hordein storage proteins[27].

Ultimately, time and cost determine the feasibility of the current and alternative DUS methods. Here, we evaluated four methods: (1) morphological trait DUS[28], (2) speed DUS[29], (3) trait-specific marker DUS[16], and (4) genomic DUS. Among these methods, the current DUS system with morphological traits takes the longest time as it usually requires one to two years of field or glasshouse

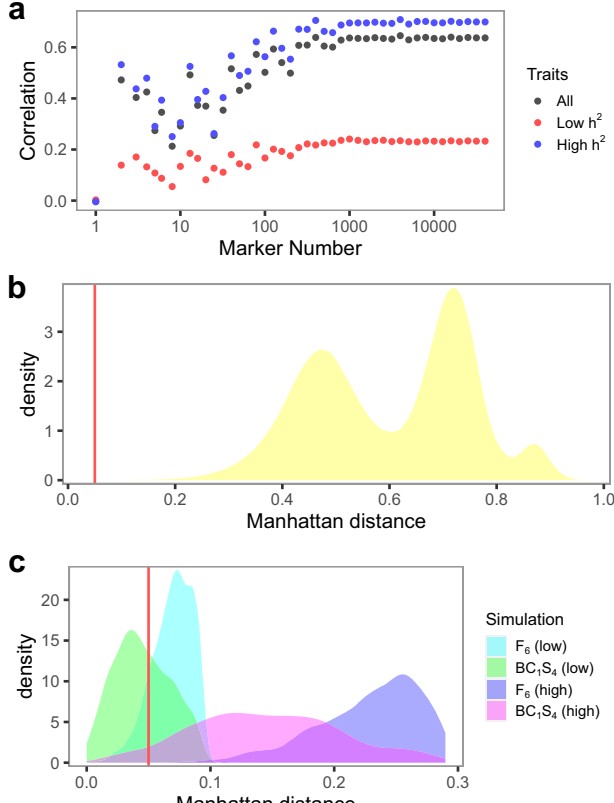

**Fig. 4 Analyses of genomic markers for DUS. a** Correlation in Manhattan distances from DUS traits and a various number of randomly sampled markers. **b** Distribution of Manhattan distances among 805 barley varieties. Within a seasonal type, distances dominate the first left peak while between seasonal type distances dominate the two right peaks. An arbitrary distance threshold of 0.05 is marked with a red vertical line. **c** Distribution of minimum Manhattan distances from each simulated progeny and reference panel (805 varieties). A similar arbitrary threshold of 0.05 is marked.

trials. Recently, Jamali et al.[29] proposed a speed DUS system by combining the current system with speed breeding[30]. While this idea is novel and interesting in regards to its purposes in DUS, it still requires experimental validation for every DUS trait since speed breeding alters plant development and many of the DUS traits are indeed developmental features. Both trait-specific marker and genomic marker methods require the least amount of time, and it is possible to shorten the time to days provided there is a routine demand. From the monetary aspect, both trait-specific and genomic marker methods cost only a small fraction of the current DUS trait method. Trait-specific markers using Kompetitive Allele-Specific PCR (KASP) cost approximately £11 for 100 markers[31] while genomic markers using the barley 50k iSelect SNP array[32] cost approximately £40 for over 40,000 markers[33]. While no cost information is available for speed DUS, it is unlikely to be less than the current DUS trait evaluation which costs £1040 per candidate variety[34]. Given all considerations, genomic markers remain the best method forward for DUS.

Being in the genomic era, we have access to great genomic resources in major crops like the barley 50k SNP array[32], wheat 90k SNP array[35], and maize 600k SNP array[36] for application in DUS. In crop species where SNP arrays are not readily available, one may consider using genotype-by-sequencing (GBS)[37] or similar methods as a starting point. As an example, we have illustrated how genomic markers can be used to evaluate the

distinctness, uniformity, and stability of new varieties (Fig. 5). Instead of relying on morphological trait differences from common knowledge varieties in the reference panel, we can determine a distance threshold based on genomic markers that would allow us to decide if a variety is sufficiently distinct. By sampling multiple seeds (or multiple pools of seeds), we can also test for uniformity based on the distances among these seeds or pools. For instance, uniformity could be defined such that the distances among the seeds from a candidate variety cannot be more than its distances with common knowledge varieties. We can quantify stability by measuring the genomic heterogeneity of the variety seed pool since a fully homogenous seed pool ensures genomic stability in subsequent generations of seed production. In an inbred species, this can be achieved by checking for genomic heterogeneity between seeds in the initial DUS application and final commercial seed lot. In an outcrossing species, this could be done by evaluating the change in allele frequencies between the initial and final seed lots after accounting for possible genomic drift. Overall, genomic markers provide a robust and effective option for improving DUS testing.

**Genomic DUS as a solution to address shortcomings in the current DUS system.** Our analysis of the current DUS system using UK barley as an example has shown that morphological traits are not fit for DUS purposes. The trait combinatorial space gets narrower over time and is likely worse in crop species with limited genetic variation. DUS traits with low heritabilities are not replicable outside the DUS trial and hence these traits have limited meaning to variety fingerprinting. As a consequence, there is no easy way for farmers to verify the identities of the varieties sown in their field. Genetic correlations between DUS and yield are detrimental to crop breeding due to the constraints imposed on selecting for higher yield and away from the common DUS trait combinatorial space. Besides, the current DUS process is time-consuming and costly, which is non-ideal for small breeding companies. Unfortunately, alternatives like trait-specific markers and small marker sets are inadequate for DUS.

It is evident that the current DUS system is due for an update as we have shown that genomic markers are the best way forward. Aside from being able to address various shortcomings in the current system, it also opens up opportunities for incorporating molecular editing into breeding systems and clarifies the boundary between new and essentially derived varieties. Given the role of the DUS system in granting varietal rights, it is the perfect setup for addressing the lack of genetic diversity in modern crops which threatens food security[38]. This, obviously, is only possible with genomic markers. In addition, with the impacts from Brexit (in the UK and EU) and Covid-19 looming for an unforeseeable future, there may be heavy restrictions on seed movement that impede the process of getting varieties into the market. Such limitations are non-ideal since only a small fraction of the candidate varieties end up passing the DUS test while the rest end up as a waste of time and money. With genomic markers for DUS, it is trivial for testing centers to either receive DNA samples from breeders or marker data from another testing center in a different country. Lastly, genomic DUS will unlock a new opportunity for an improved seed certification system to better protect breeders, farmers, and customers.

## Methods
**DUS trait and marker data.** DUS trait data from the UK national list were downloaded from the National Institute of Agricultural Botany (NIAB) and Science and Advice for the Scottish Agriculture (SASA) websites on 30th April 2020. NIAB data is available at https://www.niab.com/uploads/files/Spring_Barley_Descriptions_2019_V1.pdf and https://www.niab.com/uploads/files/Winter_Barley_Descriptions_2019_V1.pdf while SASA data is available at https://barley.agricrops.org/varieties. The NIAB data had a total of 287 barley varieties and

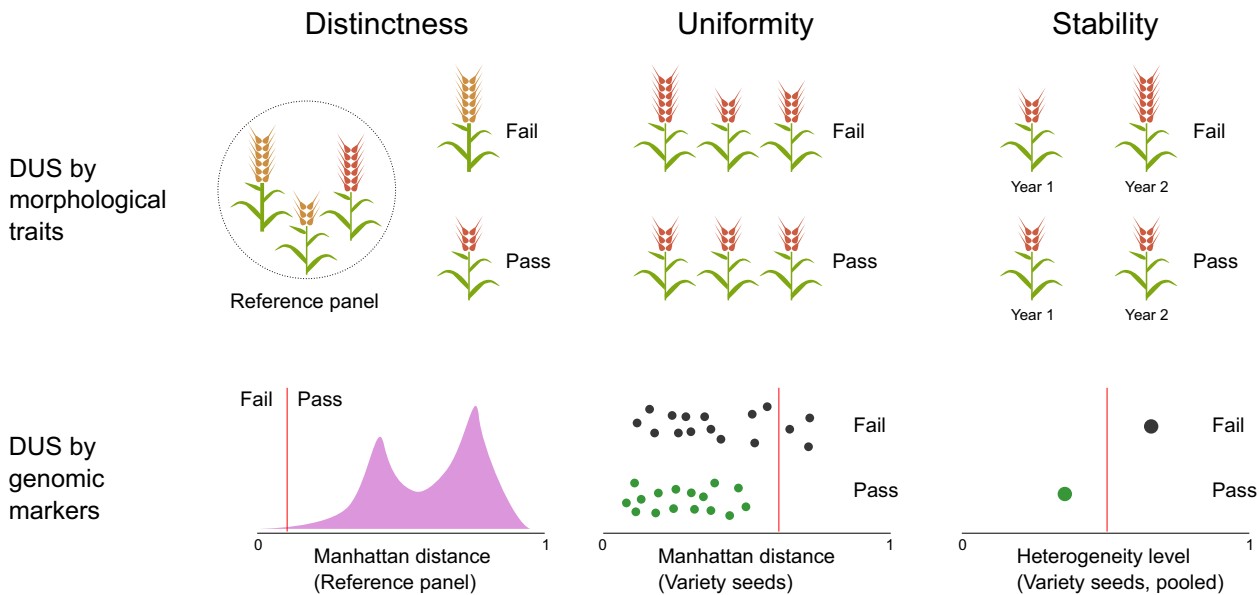

**Fig. 5 Simplified illustrations of the current DUS system and its potential successor.** In the current system, the distinctness, uniformity, and stability of a candidate variety are determined by the comparison of 28 morphological traits to a reference panel of common knowledge varieties and itself. We propose that an upgrade to the current system by using genomic markers instead of morphological traits. Here, distinctness is evaluated based on a minimum distance threshold, and anything below the threshold could be considered EDVs. Uniformity and stability evaluations are straightforward, as they each require the seeds to be close in distance to each other and low in genetic heterogeneity.

the SASA data had a total of 728 varieties. Additional DUS trait data from NIAB were taken from Cockram et al.[7], which had 577 varieties. After merging the different sources of NIAB data, there were 827 varieties remaining. The original DUS trait data were stored as text descriptions and had to be converted into numerical scales using the key provided by APHA[28]. Both NIAB and SASA data had all 28 DUS traits, although some of the traits had a high missing rate, especially in the SASA data. Therefore, we used the NIAB data for our primary analyses and the SASA data for only comparative analysis between the two. While we attempted to source as many varieties with DUS trait data as possible, we did not have an exhaustive list of all UK barley varieties to date as we were limited to those which are available publicly.

Marker data from the UK national list were obtained from the IMPROMALT project (http://www.barleyhub.org/projects/impromalt/). The original marker data contained 809 varieties with 43,799 SNP markers genotyped on the barley 50k iSelect SNP array[32]. This contrasts with a previous DUS study by Cockram et al.[7] which had 500 varieties and 1536 SNP markers. Since a large proportion of the markers did not have any missing data, we removed any marker with missing data which left us with 40,078 SNP markers. In addition, we also obtained a year of national listing and pedigree information of all varieties from the IMPROMALT project. Within these 809 varieties, 432 are spring barley, 372 are winter barley and 5 are alternative barley. We removed four varieties that did not have an Application for Protection (AFP) number, which left us with 805 varieties. Since there are only a few alternative barley varieties, we excluded them from any analysis that requires separation of the data by seasonal types. The trait and marker data were merged by their AFP numbers. Unlike the various names that are occasionally recycled, the AFP numbers are unique for each variety. They are also ordered by the date of submission for DUS testing. Overall, we had 710 varieties that are in common between the DUS trait and marker data, which serves as our primary data for analysis.

**DUS trait comparative analysis**. We calculated the DUS trait discrepancies between NIAB and SASA by taking the absolute values of the trait score differences. A total of 395 varieties were in common between the NIAB and SASA datasets. Most of the traits were scored on a scale with an increment of 1, except for traits 3, 23, and 26 which were scored as either 1 or 9. To maintain a fair comparison across all traits, we converted those trait scores from 1 or 9 to 1 or 2. All DUS trait comparisons were performed only when there is complete pairwise data between NIAB and SASA.

In addition, we subset the DUS trait data into spring and winter barley, respectively, to calculate the change in trait combinatorial space over time. This analysis was done by first sorting the barley varieties by their AFP number. Next, we computed the rolling mean of 20 varieties' Manhattan distances using the *dist* function in R[39] with an increment of one variety at a time. The lower the mean distance, the narrower the trait combinatorial space.

**Univariate mixed linear model analyses of DUS traits**. By leveraging the genomic relationship among the varieties, we partitioned the DUS phenotypic

variance into additive genetic and residual variances using *mmer* function in the "sommer" package[40] in R[39]. Briefly, the mixed model is described as:

$$y = X\beta + g + e \qquad (1)$$

For any DUS trait with *n* varieties, *y* is an $n \times 1$ vector of DUS trait, *X* is an $n \times n$ incidence matrix relating to fixed effects $\beta$, $\beta$ is an $n \times m$ matrix of *m* fixed effects, *g* is an $n \times 1$ vector of random additive genetic effect and *e* is an $n \times 1$ vector of residual effect. The *m* fixed effects included intercept, year of entry into the national listing, and seasonal type, although the last effect was dropped when spring and winter barley datasets were analyzed separately. The random additive genetic effect *g* was restricted to a normal distribution of mean 0 and variance $\sigma_g^2 A$, where $\sigma_g^2$ is the additive genetic variance and A is an $n \times n$ additive genetic relationship matrix calculated using *A.mat* function in "sommer". Similarly, the residual effect followed a normal distribution of mean 0 and variance $\sigma_e^2 I$, where $\sigma_e^2$ is the residual variance and I is an $n \times n$ identity matrix. For every DUS trait, we fitted the model using data from the spring barley dataset ($n = 370$), winter barley dataset ($n = 335$), and combined dataset ($n = 710$). We then extracted the genetic ($\sigma_g^2$) and phenotypic ($\sigma_y^2$) variances and calculated heritabilities ($h^2$) as follows:

$$\sigma_y^2 = \sigma_g^2 + \sigma_e^2 \qquad (2)$$

$$h^2 = \frac{\sigma_g^2}{\sigma_g^2 + \sigma_e^2} \qquad (3)$$

**Calculating best linear unbiased estimates (BLUEs) for yield**. We obtained the raw dry matter yield data for spring barley from Mackay et al.[23] and the Agriculture and Horticulture Development Board (AHDB) website for 509 varieties that were included in the VCU trials from 1948 to 2019. These varieties were trialed in multiple environments and years. The dry matter yield data from 1983 and onwards were taken from fungicide-treated trials, and the data prior to that were taken from "best local practice" trials which meant that fungicide usage was left to the discretion of managers at each trial. To account for this difference, we created a "management" variable. Varieties from 1983 and onwards were scored as 1 and the varieties prior to that were scored as 0 for this variable.

The raw dry matter yield data were fitted into a mixed linear model using *lmer* function in the "lme4" package[41] in R[39]. Briefly, the raw dry matter yield was set as the response variable, with variety as fixed effects, and management, management-by-year, management-by-year-by-variety, and management-by-year-by-location as random effects. Next, we calculated the best linear unbiased estimates (BLUEs) for yield using the *emmeans* function in "emmeans" package[42] in R[39].

**Bivariate mixed linear model analyses of DUS traits and yield**. We merged the DUS traits and yield data by the variety AFP numbers, which left us with 192 spring barley varieties in common. Unfortunately, we did not have access to older winter barley dry matter yield data, so the analysis here is limited to spring barley. Similar to the univariate analyses, we fitted each DUS trait and dry matter

yield BLUE into a mixed linear model using *mmer* function in "sommer" package[40] in R[39]. Briefly, the bivariate models can be written as the following:

$$y_1 = X\beta + g_1 + e_1 \tag{4}$$

$$y_2 = X\beta + g_2 + e_2 \tag{5}$$

For any pair of DUS trait and yield with *n* varieties, $y_1$ is an $n \times 1$ vector of DUS trait, $y_2$ is an $n \times 1$ vector of yield, *X* is an $n \times n$ incidence matrix relating to fixed effects $\beta$, $\beta$ is an $n \times m$ matrix of *m* fixed effects, $g_1$ is an $n \times 1$ vector of random additive genetic effect for DUS trait, $g_2$ is an $n \times 1$ vector of random additive genetic effect for yield, $e_1$ is an $n \times 1$ vector of residual effect for DUS trait and $e_2$ is an $n \times 1$ vector of residual effect for yield. The *m* fixed effects included intercept and year of entry into the national listing. Unlike the univariate analyses, here the random additive genetic effect $g_1$ and $g_2$ were restricted to a multivariate normal distribution of mean 0 and variance $\begin{vmatrix} \sigma_{g1}^2 & \rho_g\sigma_{g1}\sigma_{g2} \\ \rho_g\sigma_{g1}\sigma_{g2} & \sigma_{g2}^2 \end{vmatrix} \otimes A$, where $\sigma_{g1}^2$ is the additive genetic variance for DUS trait, $\sigma_{g2}^2$ is the additive genetic variance for yield, $\rho_g$ is the additive genetic correlation between DUS trait and yield, $\otimes$ is a Kronecker product and A is an $n \times n$ additive genetic relationship matrix calculated using *A. mat* function in "sommer". Similarly, the residual effect followed a multivariate normal distribution of mean 0 and variance $\begin{vmatrix} \sigma_{e1}^2 & \rho_e\sigma_{e1}\sigma_{e2} \\ \rho_e\sigma_{e1}\sigma_{e2} & \sigma_{e2}^2 \end{vmatrix} \otimes I$, where $\sigma_{e1}^2$ is the residual variance for DUS trait, $\sigma_{e2}^2$ is the residual variance for yield, $\rho_e$ is the residual correlation between DUS trait and yield and I is an $n \times n$ identity matrix. From the bivariate mixed models, we extracted the genetic correlation as $\rho_g$ and phenotypic correlation as $\rho_y$ where:

$$\rho_y = \frac{\rho_g\sigma_{g1}\sigma_{g2} + \rho_e\sigma_{e1}\sigma_{e2}}{\sqrt{\left(\sigma_{g1}^2 + \sigma_{e1}^2\right)\left(\sigma_{g2}^2 + \sigma_{e2}^2\right)}} \tag{6}$$

**GWAS on DUS traits**. We performed GWAS on each DUS trait using data from the spring barley dataset ($n = 370$), winter barley dataset ($n = 335$), and combined dataset ($n = 710$). We used a similar model as the univariate mixed linear model for GWAS as provided by the *GWAS* function in "sommer" package[40] in R[39]. Briefly, the GWAS model is described as below:

$$y = X\beta + m_ik_i + g + e \tag{7}$$

For any trait *y*, $m_i$ is an $n \times 1$ vector of marker genotype, $k_i$ is the marker effect and *i* is the marker index from one to the total number of markers. The other terms are the same as previously described in Eq. 1. We evaluated the GWAS results for significant markers by using a threshold of false discovery rate (FDR) of 0.05, as determined from *qvalue* function in "qvalue" package[43] in R[39]. Since barley is an inbreeding species, linkage disequilibrium (LD) can complicate GWAS results especially when there is a highly significant marker. Therefore, for any trait where the marker significance exceeded $-\log_{10}p$ of 10, we performed a follow-up GWAS with the most significant marker as a fixed effect. The re-evaluation threshold was chosen as 10 to minimize the number of GWAS runs as we were only interested in identifying any potential peaks that are masked due to major segregating loci. If any of the markers on other chromosomes were initially significant due to LD with the causative locus, then these markers should drop below the significance threshold in the second GWAS.

**Evaluation on the usefulness of small marker set in DUS via simulation**. To evaluate the 45 DUS markers in Owen et al.[25], we simulated these markers in the progeny of known parent pairs. We used 39 out of the 45 markers for simulation as six of the markers were either absent or low quality in our dataset. Based on the pedigree information, there were 212 varieties with marker data available for their parents and these varieties were generated from an intercross between the parents. For each variety and its parents, we simulated 10,000 $F_6$ progeny using "Alpha-SimR" package[44] in R[39]. We then compared the simulated progeny to the known progeny (variety) and its two parents and counted the number of exact matches in the DUS markers. In addition, we bootstrapped the comparisons 1000 times to get a better estimate of the mean count of exact matches. For comparison within the simulated progeny, we tabulated the number of occurrences of each progeny with a unique DUS marker haplotype.

**Comparing Manhattan distances from DUS traits against different number of markers**. To evaluate the number of markers needed for DUS, we randomly sampled one to the maximum number of markers with an increment of $\log_{10}$ of 0.1. We then calculated the Manhattan distances from DUS traits and markers using *dist* function in R[39]. For each set of markers, we computed the correlation between the Manhattan distances from DUS traits and marker data. In addition, we also separated the DUS traits into a high heritability group ($h^2 > 0.5$) and low heritability group ($h^2 < 0.5$), and computed the correlations similarly.

**Demonstrating the use of genomic markers in DUS via simulation**. To test how genomic markers can be used in DUS, we chose two known spring barley parent pairs with low and high genomic distances. Acumen's parents, Propino and Quench with a distance of 0.20 represents the low distance option, while Berwick's parents, Riviera and Cooper with a distance of 0.59 represent the high distance option. From each of these parent pairs, we simulated 1000 $F_6$ and $BC_1S_4$ progeny using the "AlphaSimR" package[44] in R[39]. We then computed the Manhattan distances from each simulated progeny group using *dist* function in R[39].

**Reporting summary**. Further information on research design is available in the Nature Research Reporting Summary linked to this article.

## Data availability

The compiled DUS trait data from NIAB and SASA and BLUEs for dry matter yield are available in Supplementary Data 3, 4, and 5, respectively. The IMPROMALT marker data is available at http://www.barleyhub.org/projects/impromalt/ subject to permission from the James Hutton Institute. All source file website links have been archived at https://web.archive.org/.

## Code availability

The custom R scripts for all analyses are available at https://github.com/cjyang-sruc/DUS. R version 4.0.2 was used and the associated R packages include reshape2_1.4.4, ggplot2_3.3.2, ggrepel_0.8.2, lme4_1.1-23, emmeans_1.5.2-1, sommer_4.1.1, AlphaSimR_0.12.2 and qvalue_2.20.0.

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

## Acknowledgements

We thank the IMPROMALT consortium, especially Paul Shaw, Hazel Bull, and Malcolm Macaulay, for sharing the marker data for all barley varieties analyzed here, as well as Rajiv Sharma, Ian Dawson, and David Marshall for helpful discussion throughout the work.

## Author contributions

W.P. and I.M. proposed the research idea. C.J.Y. performed the data analysis. J.R., L.R., and W.T. provided the marker data. C.J.Y., W.P., and I.M. wrote and revised the manuscript. All authors read and approved the manuscript.

## Competing interests

The authors declare no competing interests.
