## [Peer Review File · Communications Biology]

Reviewers' comments:

Reviewer #1 (Remarks to the Author):

Line 1. Insert the word .."Plant".. before the word.. "varieties"

Line 25. UPOV has provided guidelines .. in UPOV/EXN/EDV/2 of 2017 on DUS for EDVs based on molecular testing, as well as Guidelines for New types of characteristics .. TGP/15. The sentence can be modified to reflect this.

Line 20. Modify phrase. There have been several changes at UPOV to accommodate non-morphological characteristics in TGP1/3 and UPOV/ED/2.

Line 34. It is not clear the authors are in agreement with condition provided in the UPOV's guideline or are advancing an alternative approach as advance line 59-61. The phrase can be reworded for clarity.

Line 50. and Line 189. Definitions of new varieties and EDVs is provided in TGP 14, TGP1/3, UPOV/EXN/ED/2. The sentence can be modified

Line 73. There is a missing word, Are scored for.... using.

Line 79. Provide more clarity on Average Consistency. Does it mean similar scores or scoring within a range and how the range is determined to arrive at the conclusion of consistency between agencies.

Line 81 and 88. "Small" should be fixed to a statistical significance to show meaningful difference.

Line 91. The inference arising from the Manhattan distances can apply to the study area but may not be adequate to make inferences about cross-country since the analysis did not compare cross country variation.

Line 95. Can be modified. The Inference that two thirds consistency across organisations is inadequate and risky is probably not based on a benchmark unless it can be cited. It is also not in harmony with information provided in line 81, that the differences are expected.

Line 99 quotes data provided and concludes that low heritability leads to low replicability. Any available data to show this trend can be cited even from existing literature.

Line 191. The phrase may not be correct for all possible locations and can be modified.

Line 194 . The sentence can be modified to focus on DUS unless the authors can show the effect of VCU on the current scope of the study

Line 247. Remove the word... 'and'... appearing after the word.... "Update"..

Line 249. Consider revising the phrase for clarity.... from the word... 'bringing' to the word... 'practices'.

Line 251. The phrase is distantly related to the study and can be removed without losing the value of the study.

Reviewer #2 (Remarks to the Author):

Review: 6669_0

The authors address a critical and universal issue of new variety registration. Using barley DUS data, they first evaluated the commonly used method of phenotypic traits, then compared it with the use of few markers associated with DUS traits and the use of genomic markers. They briefly discussed the speed breeding method and compared the four methods in terms of time and cost. And conclude that genomic markers are more suitable than the four methods.

The manuscript is well written, easy to read. The statistical methods are well explained and applied. The conclusions are in line with the results.

Below are some remarks:

- The text has a number of missing text and typos (e.g. line 63: extra parentheses, line 71: missing word (data from....))
- Figure S2: traits not clear.
- Part of the results reads like discussion. The authors should consider moving the last 2 paragraphs of the results to the discussion section.
- Speed DUS is barely talked about.
- The discussion section is rushed. I was expecting to find something generalized beyond barley. For instance, what happens to crops which don't have a SNP array that produces reproducible markers? What about some species which show no dissimilarity using molecular markers, whose diversity depends solely on epigenetics?

Reviewer 1

	Comments	Responses
1	Line 1. Insert the word .."Plant".. before the word.. "varieties"	Done.
2	Line 25. UPOV has provided guidelines .. in UPOV/EXN/EDV/2 of 2017 on DUS for EDVs based on molecular testing, as well as Guidelines for New types of characteristics .. TGP/15. The sentence can be modified to reflect this.	Done, we have included these references in the introduction section.
3	Line 20. Modify phrase. There have been several changes at UPOV to accommodate non-morphological characteristics in TGP1/3 and UPOV/ED/2.	Done, we have modified the phrase and cited the appropriate documents in the introduction section.
4	Line 34. It is not clear the authors are in agreement with condition provided in the UPOV's guideline or are advancing an alternative approach as advance line 59-61. The phrase can be reworded for clarity.	Done, the statement in Line 34 is a neutral description of the current DUS system. The statement in Line 59-61 has been reworded for clarity.
5	Line 50. and Line 189. Definitions of new varieties and EDVs is provided in TGP 14, TGP1/3, UPOV/EXN/ED/2. The sentence can be modified	Done, we have added the UPOV references and rephrased the sentences.
6	Line 73. There is a missing word, Are scored for.... using.	Done, we have edited the sentence.
7	Line 79. Provide more clarity on Average Consistency. Does it mean similar scores or scoring within a range and how the range is determined to arrive at the conclusion of consistency between agencies.	Done, we have added new sentences for additional clarity on average/overall consistency.
8	Line 81 and 88. "Small" should be fixed to a statistical significance to show meaningful difference.	Done, we have added the mean and t-test results here.
9	Line 91. The inference arising from the Manhattan distances can apply to the study area but may not be adequate to make inferences about cross-country since the analysis did not compare cross country variation.	The DUS trait data were taken from NIAB (England) and SASA (Scotland), which are technically two countries with devolved administrations. We think our inference is conservative. It is fair to assume larger variation in DUS traits when comparing UK to elsewhere (e.g. Italy, Turkey, Australia).
10	Line 95. Can be modified. The Inference that two thirds consistency across organisations is inadequate and risky is probably not based on a benchmark unless it can be cited. It is also not in harmony with information provided in line 81, that the differences are expected.	Our findings are novel, and our inference is fair given that the DUS trait data used here were scored by trained professionals in the respective organisations. The conclusion here agrees with biological expectation – environment influences trait expression.
11	Line 99 quotes data provided and concludes that low heritability leads to low replicability. Any available data to show this trend can be cited even from existing literature.	Done, we have clarified the statement and added the appropriate reference.
12	Line 191. The phrase may not be correct for all possible locations and can be modified.	Done, we have rephrased the sentence.

13	Line 194 . The sentence can be modified to focus on DUS unless the authors can show the effect of VCU on the current scope of the study.	Done, we have removed the statement on VCU here.
14	Line 247. Remove the word... 'and'... appearing after the word... "Update"..	Done, we have edited this accordingly.
15	Line 249. Consider revising the phrase for clarity.... from the word... 'bringing' to the word... 'practices'.	Done, we have corrected the phrase.
16	Line 251. The phrase is distantly related to the study and can be removed without losing the value of the study.	Done, we have removed the sentence.

Reviewer 2

	Comments	Responses
1	The text has a number of missing text and typos (e.g. line 63: extra parentheses, line 71: missing word (data from....))	Done, we have added the missing word.
2	Figure S2: traits not clear.	Done, we have added the trait names.
3	Part of the results reads like discussion. The authors should consider moving the last 2 paragraphs of the results to the discussion section.	Done, we have merged the results and discussion together as per Communication Biology guideline. This was initially formatted for Nature Plants prior to the transfer.
4	Speed DUS is barely talked about.	Done, we have clarified more on speed DUS. It is a really interesting idea, and we are keen to see the empirical evidence for speed DUS in the near future.
5	The discussion section is rushed. I was expecting to find something generalized beyond barley. For instance, what happens to crops which don't have a SNP array that produces reproducible markers?	Done, we have addressed the use of genomic DUS in other crop species and crops that do not have SNP arrays.
6	What about some species which show no dissimilarity using molecular markers, whose diversity depends solely on epigenetics?	We are not aware of any literature that describes a new crop variety being derived from solely epigenetics. It would be an issue for genomic DUS if there is such case, but that would not work in the current DUS system anyway. It is tricky to evaluate epigenetics in terms of crop variety since the epigenetic marks can be stable for either short or long term. It is an interesting question that is worth investigating in the future, but for now, discussion on epigenetics would deviate from the messages and goals in this manuscript.